# Evaluation of Methylene Blue Migration from Time—Temperature Indicators Using LC-MS/MS

**DOI:** 10.3390/foods13121888

**Published:** 2024-06-15

**Authors:** Seung-Eun Lee, Kyung-Jik Lim, Yoon-Hee Lee, Han-Seung Shin

**Affiliations:** 1Department of Food and Medical Product Regulatory Policy, Dongguk University, 32, Dongguk-ro, Ilsandong-gu, Goyang-si 10326, Gyeonggi-do, Republic of Korea; lsx1369@naver.com; 2Department of Food Science and Biotechnology, Dongguk University, 32, Dongguk-ro, Ilsandong-gu, Goyang-si 10326, Gyeonggi-do, Republic of Korea; kyung9209@naver.com (K.-J.L.); dldbsgml491@naver.com (Y.-H.L.)

**Keywords:** smart packaging, intelligent packaging, time–temperature indicator, methylene blue, liquid chromatography–mass spectrometry/mass spectrometry (LC-MS/MS)

## Abstract

The purpose of this study was to evaluate and validate methylene blue migration from printed time–temperature indicators (TTIs) into food. It also highlights the importance of establishing regulatory measures and safety standards for food packaging, suggesting that this can contribute to improving food packaging safety. Liquid chromatography–mass spectrometry (LC-MS/MS) was used to quantify methylene blue migration in various food simulant and food matrix samples. The results show that the level of methylene blue migration varies significantly depending on the chemical properties of the food mimetic and the composition of the food matrix. The established method demonstrated a high sensitivity, with limits of detection (LODs) of 0.0019–0.0706 μg/L (kg) and limits of quantification (LOQs) of 0.0057–0.2138 μg/L (kg). This study highlights the need for a regulatory framework to mitigate the health risks associated with methylene blue in intelligent packaging systems and argues that regulatory thresholds should be set to ensure food safety and quality.

## 1. Introduction

Smart packaging integrates both intelligent and active packaging, which collectively monitor internal and external alterations within products, responding actively through interactions with external systems [1]. While active packaging targets the preservation and enhancement of food quality, safety, and sensory attributes, intelligent packaging identifies specific food matrix characteristics, offering insights into the food’s history, quality, and safety [2].

Time–temperature indicators (TTIs) represent cost-effective, irreversible tools for monitoring food quality and are renowned for their consumer-friendly nature [3]. A printed TTI consists of a material part and a film part. Methylene blue, which is harmful to humans, is used in the material part.

Methylene blue (the structure of which is depicted in Figure 1) is a heterocyclic aromatic compound characterized by a blue color and is hydrophilic [4,5]. This basic dye has a significant heterocyclic aromatic structure and is widely used in various fields, including industry, chemistry, biology, and medicine [6]. For instance, it is utilized in the manufacturing of color pens, printing inks, and other coloring industrial products. Despite its widespread use, methylene blue has been associated with potential toxicity and adverse effects in humans [7]. Small doses of methylene blue, previously considered safe, can precipitate serotonin toxicity [8]. Also, clinical trials have shown that acute exposure to methylene blue can lead to adverse effects including an increased heart rate, vomiting, shock, cyanosis, jaundice, paralysis, and tissue necrosis. Nadler et al. found that intravenous administration of high doses of methylene blue to adult males and Caucasian male infants resulted in various side effects [9]. Additionally, Lee and Wurster revealed that methylene blue could induce cytotoxicity in certain human cells [10].

The medical use of methylene blue is no longer permitted in some countries such as Switzerland [6]. The United States Food and Drug Administration (FDA) guidelines specify contraindications for the concomitant use of methylene blue with certain medications [11]. The European Union, for example, has implemented an outright ban on the use of methylene blue in animals intended for food production, aiming to prevent chemical residues from contaminating the human food supply [12]. Conversely, in Japan, the approach is to manage its use through setting a maximum residue limit of 10 μg/kg in seafood, reflecting a more controlled utilization of this compound in food safety [13]. These variations in regulatory frameworks illustrate the global complexity of managing methylene blue’s application, balancing its therapeutic benefits against potential health and safety risks.

Given the potential health risks associated with methylene blue, particularly its migration from TTIs to food, there is a critical need for comprehensive research to assess the extent of migration, the factors influencing it, and the potential health implications for consumers. While the migration kinetics and toxicological effects of methylene blue have been extensively studied [7,8], knowledge remains limited regarding the quantification of methylene blue transferred from food packaging, specifically from TTIs, to food products.

Migration analysis is a widely utilized method for assessing the transfer of substances from packaging materials to food. Prior research has extensively documented various factors influencing migration, including the contact time, the temperature, and the use of food simulants. Additionally, the limitations associated with using simulants instead of real food have been highlighted, suggesting that real food provides more accurate migration data [14,15,16]. This study seeks to address these gaps and articulate its contributions to the regulatory framework more clearly.

Analyzing methylene blue in food matrix group samples requires chemical preparation methods to extract and isolate the methylene blue from complex matrix samples [17]. Carrez solution, HybridSPE^®^-Phospholipid cartridges, and Bond Elut Alumina A cartridges were utilized for the purification of the separated extracts [17,18,19,20,21,22]. Liquid chromatography–mass spectrometry/mass spectrometry (LC-MS/MS) has emerged as an analytical tool for the analysis of methylene blue in food matrix group samples due to its high sensitivity, selectivity, and capability for multi-component analysis [23]. LC-MS/MS offers advantages such as high-throughput analysis, low detection limits, and reliable quantification, making it ideal for monitoring trace levels of methylene blue in complex food samples.

Therefore, this study seeks to explore the migration of methylene blue from food packaging to food, highlighting the importance of developing regulatory measures and safety standards for food packaging. This analysis highlights the necessity for regulatory measures and enhancements in food packaging safety. Furthermore, this study applied an analytical method tailored to the food samples, suggesting that its application could aid in establishing regulations for methylene blue and contribute to improvements in food packaging safety.

## 2. Materials and Methods

### 2.1. Reagents and Materials

The hazardous material standard, including methylene blue, was purchased from Sigma-Aldrich, Co. (Darmstadt, Germany). Acetic acid was also purchased from Sigma-Aldrich, Co. (Darmstadt, Germany). All reagents were of analytical grade, with n-heptane being sourced from Daejung (Seoul, Republic of Korea), whereas ethanol, HPLC-grade water, and acetonitrile were obtained from Samchun (Seoul, Republic of Korea). Polytetrafluoroethylene (PTFE) syringe filters with a pore size of 0.45 μm were purchased from Whatman (Seoul, Republic of Korea). Two types of solid-phase extraction (SPE) were used to purify methylene blue from the samples. A HybridSPE^®^-Phospholipid cartridge (30 mg, 1 mL) was purchased from Sigma-Aldrich, Co. (Darmstadt, Germany) and a Bond Elut Alumina A cartridge (500 mg, 3 mL) was purchased from Agilent (Santa Clara, CA, USA). Five kinds of food samples, including liquid egg, beef, mackerel, almond milk, and potato, were used. Liquid egg was purchased from Pulim Food (Jincheon-gun, Chungcheong Buk-do, Republic of Korea). Beef and potato were purchased from Mamma Agricultural Mart in Ilsan, Gyeonggi-do, Republic of Korea. Mackerel was purchased from Andong mackerel (Andong, Gyeongsangbuk-do, Republic of Korea) and almond milk was purchased from Maeil Dairies Co., Ltd. (Seoul, Republic of Korea).

### 2.2. Selection of Samples

Sample selection was categorized into two main groups: a food simulant group and a food matrix group. Food simulants are reagents with properties similar to those of food, allowing them to be used as substitutes. Each country has specific food simulant requirements based on the characteristics of different foods. In the US, 10% ethanol is used for aqueous and acidic foods, 10% and 50% ethanol for alcoholic foods, and cooking oil for fatty foods [24]. In the EU, 10% ethanol is used for hydrophilic foods, while 3% acetic acid is used for hydrophilic foods with a pH less than 4.5. More lipophilic alcoholic foods require 20% ethanol, and 50% ethanol is used for lipophilic/alcoholic foods (oil-in-water emulsion). For lipophilic foods, vegetable oil serves as the food simulant [25]. Table 1 details the food simulants stipulated by the Ministry of Food and Drug Safety in Korea, which include n-heptane, 20% ethanol, 50% ethanol, water, and 4% acetic acid [26]. These simulants are designated for specific food groups: oil and fatty foods with an oil content of 20% or more correspond to n-heptane, foods with an alcohol content of 20% or less correspond to 20% ethanol, and those with an alcohol content of 20% or more correspond to 50% ethanol. Foods with a pH of 5 or less correspond to 4% acetic acid, and those with a pH greater than 5 correspond to water. These five food simulants were used in this experiment.

The food matrix group was divided into a total of four groups. The standards for dividing the matrix in this study were, firstly, into liquid and solid foods, and secondly, into whether the protein content was more than or less than 11% [27].

#### 2.2.1. Sample Preparation for the Food Simulant Group and Food Matrix Group

For the elution test, 10 mL of acetonitrile (ACN) was added to 2 mL of sample. The sample was then vortexed and filtered through a 0.4 μm polytetrafluorethylene (PTFE) membrane. The filtered solution was then analyzed via LC-MS/MS [28].

For low-protein liquid and solid transfer tests, 10 mL of ACN was added to 2 g of sample. The obtained solution was then centrifuged at 8000× *g* for 10 min after removing the sample. After collecting the supernatant, 10 mL of ACN was added and the sample was once again centrifuged at 8000× *g* for 20 min. Next, 5 mL of the supernatant was placed in a HybridSPE^®^-Phospholipid cartridge previously activated with 5 mL of ACN and washed with ACN. Finally, the sample was filtered with PTFE filters and analyzed via LC-MS/MS [17].

For high protein liquid and solids transfer test, 10 mL of ACN was added to 2 g of the sample. Following sonication, the mixture was incubated at 80 °C for 20 min. After removing the sample, 2 mL of Carrez I solution and 2 mL of Carrez II solution were added, respectively, and vortexed [19,21]. After centrifuging at 8000× *g* for 20 min, 5 mL of the supernatant was transferred to a Bond Elut Alumina A SPE cartridge previously activated with 5 mL of ACN and washed with ACN. Subsequently, 1 mL of a solution of ammonium acetate buffer acidified with 0.2% formic acid in ACN was dissolved. Finally, the solution was filtered using PTFE filters and analyzed using LC-MS/MS [17].

The clarity and reproducibility of the methylene blue analysis method used in this study were improved by referring to the method from the study by Zhang et al. and the LC-MS/MS method for methylene blue analysis used in various other studies [17,28]. For accurate analysis of methylene blue, Carrez reagent was utilized to precipitate proteins, ensuring minimal interference from protein matrices during the LC-MS/MS analysis. The use of a HybridSPE^®^-Phospholipid cartridge and a Bond Elut Alumina A SPE cartridge further purified the samples by selectively removing phospholipids and other potential contaminants [17,18,19,20,21,22].

#### 2.2.2. Methodology for Elution, Transfer, and Full-Scale Transfer Analyses for Samples

In the conducted study, the migration of methylene blue from the TTI into a designated food simulant group was assessed through an elution test. Then, a transfer test was conducted to evaluate the extent to which methylene blue was transferred from the TTI into the actual food matrix. Furthermore, an additional study was conducted to quantify the migration of methylene blue into the food matrix group under conditions of physical damage, simulating real-world scenarios, where TTIs may be compromised. This methodology was selected to reliably assess potential contamination across diverse conditions.

This experiment was conducted to assess the amount of methylene blue migration from three types of TTIs, enzymatic TTIs, printed TTIs, and Vitsab’s TTIs, into food samples. The experiment was designed to evaluate the exposure of a TTI with a surface area of 1 cm^2^ to 2 milliliters of food simulant. This setup facilitated the assessment of the TTI’s interaction with the simulant under controlled conditions. The food simulant was then heated to a temperature of 70 °C. After heating, the food simulant was introduced into a Teflon gasket cell, with the TTI was oriented downward to maximize exposure to the simulant. This assembly was placed within a dry oven preheated to 70 °C. Depending on the type of food simulant used—water, 4% acetic acid, 20% ethanol, or 50% ethanol—the Teflon gasket remained in the oven for a duration of 30 min. For experiments involving n-heptane as the food simulant, the setup was adjusted to a 25 °C dry oven, where it was maintained for one hour [26]. These temperature and time settings were established with reference to international standards from the FDA and EFSA. According to EFSA guidelines, the European Union regulation (EU No 10/2011) specifies various test conditions to evaluate the migration of substances from packaging materials into food [29]. Additionally, the n-heptane extraction conditions were set according to FDA regulations, with a temperature of 25 °C for 30 min to assess the migration potential of substances from packaging materials under short-term contact at room temperature [30]. This study was conducted following the guidelines set by the Ministry of Food and Drug Safety in Korea, with reference to FDA and EFSA international standards to scientifically evaluate the safety of packaging materials. Following these experimental procedures, additional steps were undertaken with the test solution, as outlined in Section 2.2.1 of the study protocol.

When conducting transfer tests and full-scale transfer tests, the TTI was placed in a sealed container with 30 g of the sample and then stored in a dry oven at 40 °C for 10 days [24]. After the preparation, TTIs stored with the food matrices were used as the test sample for further analysis. The test samples underwent additional processing, as outlined in Section 2.2.1.

### 2.3. Instrumentation

Methylene blue was systematically analyzed using liquid chromatography–tandem mass spectrometry (LC-MS/MS), a sophisticated analytical technique renowned for its high sensitivity and specificity. This method involves the separation of methylene blue molecules via liquid chromatography (LC) followed by their detection and quantification using mass spectrometry (MS). The tandem mass spectrometry (MS/MS) configuration allows for further fragmentation of the molecules, providing unique spectral data that facilitate the detailed characterization of a compound’s chemical structure and properties. LC-MS/MS coupled with an AB Sciex triple quadrupole mass spectrometer (AB SCIEX API3200) was employed. A C18 column (50 mm × 2.1 mm, 2.6 μm, Phenomenex, Torrance, CA, USA) operated at a constant temperature of 40 °C was applied for analyte separation. Mobile phase A consisted of HPLC-grade 0.1% formic acid, while mobile phase B consisted of acetonitrile containing 0.1% formic acid [31]. The initial composition began from 90% A to 10% B. This composition was maintained constant for the first 2 min. A linear gradient was then applied, where the composition was adjusted to 10% A and 90% B over a 3 min period, reaching this ratio at 5 min. The composition of 10% A to 90% B was maintained until 7.9 min. A rapid change in the mobile phase composition was then initiated, returning to 90% A and 10% B within 0.1 min, and this composition was held for the remaining duration of the run until 10 min. The flow rate was set as 0.3 mL/min.

The mass spectrometer utilized electrospray ionization (ESI) in the positive ion mode with a spray voltage of −5500 V and a vaporizer temperature of 450 °C. Under these conditions, no sample contamination or sample overlapping was observed. The instrument was operated in multiple reaction monitoring (MRM) mode. The collision energies were optimized for individual analytes. The “compound-dependent” parameters—declustering potential (DP), entrance potential (EP), collision energy (CE), collision cell entrance potential (CEP), and collision cell exit potential (CXP)—for the API 3200 MS/MS system are presented in Table 2 for methylene blue. Declustering potential (DP) refers to the voltage applied to break up clusters of ions, entrance potential (EP) is the voltage that helps ions enter the mass spectrometer, collision energy (CE) is the energy used to fragment the ions, and collision cell entrance potential (CEP) and collision cell exit potential (CXP) are voltages applied to control the movement of ions into and out of the collision cell, respectively. The retention time for methylene blue was 7.02 min, as shown in Figure 2 and Table 2. The mass spectrum of methylene blue is shown in Figure 3.

### 2.4. Method Validation

The LC-MS/MS analytical method was validated for linearity (R^2^), accuracy (%), relative standard deviation (RSD) (%), limits of detection (LODs), and limits of quantification (LOQs) on both food simulant group samples and food matrix group samples. Linearity assessment was conducted by using least square linear regression, analyzing the calibration curves for each element, and calculating the slope, intercept, and coefficient of determination [32,33].

Accuracy and precision were evaluated by analyzing diluted standard solutions spiked with the analytes. The calibration curves for all analytes were generated using five different concentrations. Methylene blue analysis was conducted using standard mixtures at concentrations of 0.10, 0.50, 1.00, 5.00, and 10.00 µg/L(kg). All analyses adhered to CODEX guidelines [34]. To assess accuracy, the calibration curve was evaluated at five different concentrations with three repetitions for both intra-day and inter-day measurements. Accuracy was then calculated using Equation (1).
Accuracy (%) = (Cmean − Cblk)/Cspiked × 100 (%)(1)

In this equation, the average concentrations of the standard, blank, and spiked samples are indicated as Cmean, Cblk, and Cspiked, respectively. Precision was determined by calculating the relative standard deviation (RSD) using Equation (2).
RSD (%) = Csd/Cmean × 100 (%)(2)

In Equation (2), Csd represents the standard deviation of the concentration.

The LOD and LOQ values were, respectively, determined as the concentrations corresponding to 3.3 and 10 times the standard deviation of the blank solution measurements divided by the slope of the calibration curves, as defined in Equation (3).
LOD = 3.3 × σ/S, LOQ = 10 × σ/S (3)

### 2.5. Statistical Analysis

All analyses were performed in triplicate, and the data are presented as means ± standard deviation (SD). All statistical analyses were conducted using Microsoft Excel (Microsoft Corporation, Redmond, WA, USA).

## 3. Results and Discussion

### 3.1. Classification of Samples

Categorization of test conditions into “Food Simulant Group” and “Food Matrix Group” is essential for comprehensive migration studies [11,24]. Food simulant group samples enable consistent testing under standardized conditions, ensuring reproducibility and comparability across different studies. Conversely, actual food matrix group samples reflect the complex interactions and realistic use conditions that are crucial for accurate risk assessments. This dual approach ensures both reliable laboratory results and practical, real-world relevance, bridging the gap between controlled experiments and consumer safety evaluations.

Food simulants were strategically selected to replicate the interactions between various food substances and TTIs. The simulants—water, 4% acetic acid, n-heptane, 20% ethanol, and 50% ethanol—represent different food group characteristics essential for assessing the migration of methylene blue from TTIs. In the United States, 10% and 50% ethanol are used for aqueous, acidic, and alcoholic foods, while cooking oil simulates high-fat foods. The European Union employs 10% ethanol for hydrophilic foods with a pH below 4.5, 20% ethanol for lipophilic alcoholic foods, and 50% ethanol for oil-in-water emulsions, with vegetable oil for lipophilic foods.

This study employed water to simulate foods with a pH above 5, 4% acetic acid for those with a pH below 5, and n-heptane for foods high in oil and fat. Ethanol at concentrations of 20% and 50% was used to model beverages with varying alcohol contents, reflecting the distinct migration dynamics in alcoholic beverages. These choices allow for a thorough evaluation of the safety and integrity of TTIs under realistic food contact scenarios, following guidelines from the Ministry of Food and Drug Safety [26].

The study design carefully tailored the classification of food matrix group samples to the interaction between food and methylene blue in TTIs. This classification considers the chemical interactions observed with methylene blue and various proteins with complex molecular structures [9]. This study ensured a precise analysis of methylene blue migration from TTIs to food; the samples were divided based on protein content into four categories: high-protein solids, high-protein liquids, low-protein solids, and low-protein liquids.

The representative foods of the four matrix group samples were mostly selected from the top 18 most consumed food according to the Korea Health Industry Development Institute’s national nutritional statistics [35]. The exceptions were almond milk and mackerel. Almond milk is categorized with soy milk, oat milk, and cashew milk as non-dietary milk. Due to its comparatively low protein content among various non-dietary milk alternatives, almond milk was selected as the sample for this study [36]. According to the Korea Maritime and Fisheries Development Institute, mackerel was chosen as the representative sample in the study due to its status as the most frequently consumed seafood among the surveyed logistics samples [37]. The division of the four food matrix group samples into high-protein and low-protein categories adhered to the standards established according to the standards of Regulation (EU) No 1169/2011 and Food and Labeling, issued by the Ministry of Food and Drug Safety in Korea [27,38]. In the case of the European Union, foods were classified as high-protein or low-protein based on their protein content per 100 g or 100 mL. Foods containing less than 5% of total calories from protein were classified as low-protein foods, while foods containing more than 20% of the NRV per 100 g or more than 10% per 100 mL were classified as high-protein or rich-protein foods. In the case of the Ministry of Food and Drug Safety in Korea, food with a protein content exceeding 11 g per 100 g is classified as high-protein food, whereas that with less than 11 g of protein per 100 g is classified as low-protein food. Details of the food simulant group samples and food matrix group samples are systematically presented in Table 2.

### 3.2. LC-MS/MS Method Validation

Method validation was conducted in accordance with CODEX guidelines [34]. The linearity equation, R^2^, accuracy, LOD, and LOQ values for methylene blue in both food simulant samples and food matrix group samples are presented in Table 3 and Table 4. All samples exhibited linearity above 0.99. The intra-day accuracy displayed a recovery range from 83.11% to 118.52%, whereas the inter-day accuracy ranged from 85.10% to 117.03%. Repeatability studies showed intra-day relative standard deviation (RSD) recoveries ranging from 0.11% to 8.67%, and inter-day RSDs ranging from 0.43% to 11.32%. The LOD values of food simulant samples (water, 4% acetic acid, n-heptane, 20% ethanol, 50% ethanol) and food matrix group samples (liquid egg, beef, mackerel, almond milk, and potato) ranged from 0.0019 to 0.0706 μg/L (kg), and LOQ values ranged from 0.0057 to 0.2138 μg/L (kg). These results confirm that the proposed analytical method adheres to CODEX guidelines and can significantly contribute to ensuring food safety and compliance with regulatory standards for food packaging.

Analysis of the LOD and LOQ values for various food simulant group samples used in the elution test showed that there was a correlation with the polarity of the solvent and its interaction with the analyte. Highly polar water displayed the lowest LOD and LOQ values, reflecting its increased interaction with the analyte and consequently its enhanced sensitivity. Conversely, less polar solvents like n-heptane demonstrated higher LOD and LOQ values, attributed to a reduced analyte solubility and weaker solvent–analyte interactions [39]. These findings underscore the significance of solvent selection in analytical methodologies and emphasize the influence of solvent polarity on detection limits. However, 4% acetic acid exhibited the highest LOD and LOQ values. This is due to its low pH affecting the results [40].

In the food matrix group samples, significant observations emerged regarding LOD and LOQ values. Solid samples such as beef, mackerel, and potato showed lower LOD and LOQ values than liquid samples. This implies that the preparation process may have had a more beneficial impact on solid samples [6]. For the preparation of food matrix group samples, a Bond Elut Alumina A SPE cartridge and Carrez reagent were used for high-protein samples, and a hybrid SPE cartridge was used for low-protein samples. The Carrez reagent process is optimal for protein precipitation and turbidity removal. Carrez reagents are crucial in enhancing analytical performance for high-protein samples [19,21]. They facilitate efficient protein separation and purification before analysis. Bond Elut Alumina A SPE cartridges have specific surface properties that enable the selective adsorption and separation of proteins and other compounds from complex matrices. This cartridge offers high recovery rates and efficient recovery of target materials. Therefore, it has proven to be an effective tool to prepare high-protein samples [17,20]. The HybridSPE^®^-phospholipid cartridge is recognized as an effective technique for preparing protein and phospholipid samples. It can selectively adsorb and purify target molecules from complex matrix samples, and this method is also efficacious in the selective extraction of methylene blue [18,22]. Therefore, appropriate protein purification using Carrez reagent, a Bond Elut Alumina A SPE cartridge, and a HybridSPE^®^-Phospholipid cartridge has emerged as an important aspect of the preparation process for analysis of methylene blue.

The objective validation method for methylene blue was implemented by both internal and external laboratories. Samples with concentrations of 0.1 μg/L(kg), 1 μg/L(kg), and 10 μg/L(kg) were tested across multiple laboratories. Applying the validated method in both internal and external laboratories yielded accuracy rates ranging from 80% to 120% for all samples, as summarized in Table 5. These findings demonstrate that the proposed analytical method complies with the guidelines established by CODEX. Therefore, through this study, it is possible to understand the movement of methylene blue from TTIs to food, and this further contributes to the understanding of the movement of hazardous substances from food packaging materials to food and the establishment of standards.

### 3.3. Elution, Transfer, and Full-Scale Transfer Test for Analysis

This study aimed to evaluate the migration of methylene blue from various types of TTIs into food, with the goal of establishing a basis for new regulations regarding acceptable levels of methylene blue in TTIs. Three types of TTIs were examined: enzymatic, printed, and Vitsab’s TTIs. The printed TTI and enzymatic TTI utilized in this investigation were manufactured in Korea, while Vitsab’s TTI was imported from Switzerland. In addition to analyzing the amount of methylene blue transferred from the printed TTI to food, three types of TTIs were used to monitor the amount of methylene blue transferred from other TTIs (enzymatic TTI, Vitsab’s TTI) to food.

Given the expected minimal risk to human health posed by trace amounts of methylene blue released from TTIs, three different tests were conducted: an elution test, a transfer test, and a full-scale transfer test. The elution test examined the migration of methylene blue within the TTIs, providing insights into the interaction between the methylene blue in TTIs and the food simulant. This evaluation assesses the effectiveness of the TTIs and measures the amount of methylene blue from the TTIs that migrates into food simulants under actual use conditions. Therefore, it can contribute to the overall evaluation of product safety. The transfer test was designed to assess the extent of methylene blue migration from the TTIs into food when the TTIs are attached to packaging. This test is crucial for evaluating potential direct methylene blue exposure in real food. The full-scale transfer test evaluated the migration of methylene blue to food when the TTIs are damaged. This test evaluates the concentration of methylene blue that migrates into the food. Before conducting the elution test, transfer test, and full-scale transfer test, control food simulant group samples and food matrix group samples were prepared equally without TTIs, and the amount of methylene blue was analyzed. Methylene blue was not detected in all samples.

The elution, transfer, and full-scale transfer test results for the printed TTI are detailed in Table 6. The results indicate that in the elution test, the minimum elution amount of methylene blue in the printed TTI was 0.06 ± 0.01 μg/L, while the maximum was about 1.20 ± 0.02 μg/L. For the transfer test, the minimum transfer amount was 3.13 ± 0.01 μg/kg, and the maximum was 6.25 ± 0.01 μg/kg. Furthermore, in the full-scale transfer test, the minimum transfer amount was 41.76 ± 0.02 mg/kg, and the maximum was 117.45 ± 0.02 mg/kg. These results indicate that the migration of methylene blue is slightly higher in the transfer test and the full-scale transfer test compared to the elution test. This can be attributed to the potential influence of factors such as the protein composition and the state of the object when applied to real food products [6,16]. Physical damage (full-scale transfer test) to printed TTIs resulted in the migration of methylene blue at concentrations as high as approximately 11.75 mg/kg, significantly exceeding the FDA’s recommended dosage of 1 mg/kg. This level also surpasses the therapeutic range of 1 to 8 mg/kg used for the treatment of serotonin syndrome [11]. These findings highlight the critical need to review and strengthen the relevant regulatory standards to ensure safety and effectiveness.

Vitsab’s TTI exhibited values ranging from 0.00 to 0.18 μg/L in the elution test, showed ND results in all transfer tests, and exhibited values ranging from 4.50 to 12.00 μg/kg in the full-scale transfer test. These results are outlined in Table 7.

The enzymatic TTI showed ND (not detected) results for all tests. Enzymatic TTIs typically do not contain methylene blue or similar indicator compounds.

Both the printed TTI and Vitsab’s TTI results exhibited detectable responses in both the elution and transfer test categories. In the elution test, quantifiable concentrations were observed for all solvents tested with the printed TTI, while Vitsab’s TTI showed quantifiable concentrations for water, 4% acetic acid, 20% ethanol, and 50% ethanol. Notably, the concentrations measured by the printed TTI tended to be higher compared to Vitsab’s TTI. This discrepancy arises from the variance in the concentration of methylene blue used in the printed TTI compared to that in Vitsab’s TTI, as well as the differences in the packaging materials of the two indicators [41].

This study elucidates the migration of methylene blue across various food simulant group samples. Among the tested food simulant group samples, methylene blue exhibited the least migration in n-heptane, while migration was most pronounced in 4% acetic acid [42]. These results indicate that migration of methylene blue is minimal when TTIs are exposed to a pH of 5 or lower. Additionally, the full-scale transfer test results showed that the migration of methylene blue was generally more pronounced in high-protein foods compared to low-protein foods. Notable differences were observed in the full-scale transfer test results, particularly concerning high-protein samples such as beef, mackerel, and liquid egg. In the transfer test category, both types of TTIs demonstrated quantifiable concentrations across various food samples, with the printed TTI generally yielding higher concentrations compared to Vitsab’s TTI.

Furthermore, this study contributes to the limited existing research on the migration of hazardous substances from TTIs to food and can aid in establishing regulations for methylene blue, thereby improving food packaging safety. However, this study has certain limitations that must be addressed. The classification of the food matrix group in this study was limited, restricting the generalizability of the findings. Therefore, additional research should be conducted to further expand the food matrix group. Moreover, beyond the printed TTI studied here, it is crucial to explore the transfer of harmful substances into food from various other developed TTIs to broaden the applicability of these findings.

## 4. Conclusions

This study demonstrated the effective establishment of a method to analyze the migration of methylene blue from printed TTIs into various food simulant and matrix group samples using LC-MS/MS, highlighting the critical influence of both the characteristics of the food simulant and the food matrix group samples’ composition on the migration levels observed. Variability in migration was discernible across different simulants and matrices, emphasizing the interaction between methylene blue and the chemical properties of the simulated food environments.

The results of this research have significant implications for food safety and intelligent packaging technologies, emphasizing the importance of establishing regulatory guidelines that effectively manage the transfer of chemicals from packaging materials into food products. This study supports the establishment of safety thresholds for methylene blue, which could guide regulatory adjustments and enhance consumer safety protocols. Specifically, the data obtained can assist in defining maximum allowable limits for methylene blue in materials in contact with food, ensuring that consumer exposure remains within safe bounds. Additionally, these findings could inform revisions of existing food packaging regulations to incorporate more stringent controls on the use of methylene blue and similar substances.

Future research should focus on long-term studies to monitor the effects of methylene blue migration over extended periods and in real-life conditions, considering factors such as temperature fluctuations and varying storage durations. Moreover, expanding the scope of research to include a broader range of food types and packaging materials will provide a more comprehensive understanding of methylene blue’s behavior in diverse scenarios. Advanced analytical techniques and interdisciplinary approaches could further refine the detection and quantification of methylene blue, contributing to the continuous improvement in food safety standards.

In conclusion, the successful development and validation of a method for analyzing methylene blue migration lay a foundational framework for future regulatory enhancements and contribute to the broader field of food safety and packaging technology. This study underscores the importance of ongoing research and regulatory oversight in the context of food safety and consumer health.

## Figures and Tables

**Figure 1 foods-13-01888-f001:**
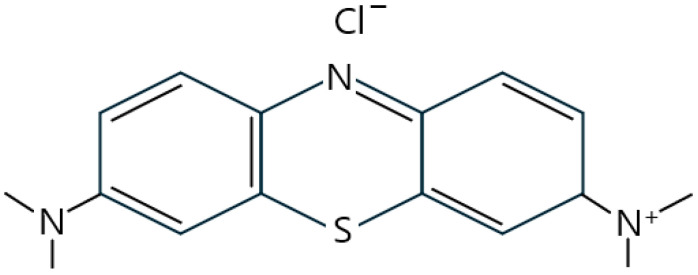
Chemical structure of methylene blue.

**Figure 2 foods-13-01888-f002:**
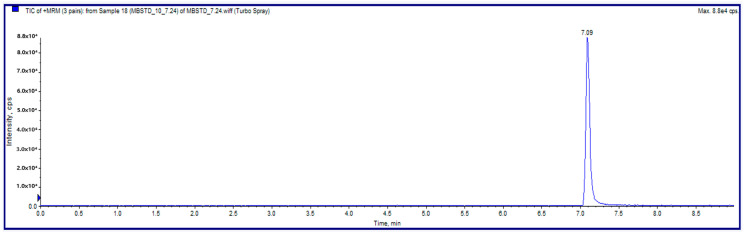
Chromatographic peaks of methylene blue.

**Figure 3 foods-13-01888-f003:**
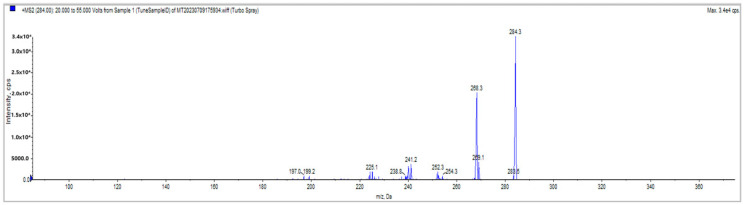
Mass spectrum of methylene blue.

**Table 1 foods-13-01888-t001:** MRM transition parameters (DP, EP, CEP, CE, and CXP) and retention times of methylene blue.

Precursor Ion (*m*/*z*)	Production (*m*/*z*)	DP ^a^ (V ^f^)	EP ^b^ (V)	CEP ^c^ (V)	CE ^d^ (V)	CXP ^e^ (V)	Retention Time (min)
284.00	268.3	58	8	18	35	4	7.02
	252.2				45		
	241.2				33		

^a^ DP: declustering potential, ^b^ EP: entrance potential, ^c^ CEP: collision cell entrance potential, ^d^ CE: collision energy, ^e^ CXP: collision cell exit potential, and ^f^ V: volt.

**Table 2 foods-13-01888-t002:** Classification of food simulant group samples and food matrix group samples.

Group	Classification	Food Type	Sample
Food simulant	pH	Food exceeding pH 5	Water
		Food with a pH of 5 or lower	4% acetic acid
	Alcoholic beverages	Alcoholic beverages containing less than 20% alcohol	20% ethanol
		Alcoholic beverages containing more than 20% alcohol	50% ethanol
	Lipid	Food containing 20% or more of lipid	n-heptane
Food matrix	Liquid	Protein content of more than 11% of food	Liquid egg
		Less than 11% of protein of food	Almond milk
	Solid	Protein content of more than 11% of food	Beef
			Mackerel
		Less than 11% of protein of food	Potato

**Table 3 foods-13-01888-t003:** Results of linearity equation, R^2^, LOD, and LOQ.

	Sample	Linearity Equation ^a^	R^2^	LOD ^b^ (μg/L (kg))	LOQ ^c^ (μg/L (kg))
Elution Test	Water	y = 28,735x − 3684.8	0.9986	0.0019	0.0057
	4% acetic acid	y = 27,352x − 1302.3	0.9998	0.0605	0.1833
	n-heptane	y = 26,736x − 3034.1	0.9994	0.0706	0.2138
	20% ethanol	y = 29,254x − 4726.5	0.9981	0.0544	0.1648
	50% ethanol	y = 27,442x − 3529.3	0.9984	0.0453	0.1372
Transfer Test	Liquid egg	y = 24,689x − 4264.4	0.9952	0.0636	0.1928
	Beef	y = 24,280x − 2526.3	0.9991	0.0197	0.0596
	Mackerel	y = 27,441x − 3526.9	0.9984	0.0453	0.1372
	Almond milk	y = 26,678x − 2869.3	0.9985	0.0632	0.1916
	Potato	y = 22,525x − 1044.6	0.9998	0.0224	0.0680

^a^ Linearity equation numbers express mean values (*n* = 3); ^b^ LOD set up at a signal-to-noise ratio (S/N) = 3.3; ^c^ LOQ set up at a signal-to-noise ratio (S/N) = 10.

**Table 4 foods-13-01888-t004:** Evaluation of accuracy and precision in food simulant group samples and food matrix group samples across intra-day and inter-day assays.

Sample	Concentration (μg/L(kg))	Intra-Day (*n* = 3)		Inter-Day (*n* = 3)	
		Accuracy (%)	RSD (%)	Accuracy (%)	RSD (%)
Water	0.1	90.10 ± 1.96	2.17	99.64 ± 4.88	9.71
	0.5	110.67 ± 1.55	1.40	111.47 ± 4.63	4.15
	1	112.52 ± 2.86	2.54	110.51 ± 1.28	1.16
	5	117.37 ± 0.67	1.25	116.22 ± 2.25	0.74
	10	106.02 ± 0.59	0.56	105.67 ± 2.16	2.05
4% acetic acid	0.1	112.44 ± 4.94	8.67	98.96 ± 3.21	6.64
	0.5	116.15 ± 0.94	0.81	112.95 ± 4.42	3.91
	1	109.70 ± 0.12	0.11	108.63 ± 0.54	4.76
	5	112.80 ± 0.94	3.76	115.56 ± 0.73	6.49
	10	100.93 ± 1.34	1.33	99.38 ± 3.15	3.17
n-heptane	0.1	100.19 ± 1.33	3.97	95.22 ± 2.42	2.54
	0.5	104.06 ± 2.03	2.07	103.97 ± 0.74	0.72
	1	98.76 ± 1.46	1.52	97.08 ± 1.66	1.71
	5	113.86 ± 1.44	1.47	111.15 ± 1.06	0.96
	10	98.20 ± 1.01	1.30	98.03 ± 1.30	1.33
20% ethanol	0.1	85.26 ± 1.05	1.24	86.52 ± 3.43	3.97
	0.5	108.48 ± 1.65	1.52	106.84 ± 3.03	2.84
	1	107.22 ± 3.74	3.48	104.72 ± 3.21	3.06
	5	118.52 ± 1.59	1.34	117.03 ± 0.50	0.43
	10	107.85 ± 0.17	0.16	106.09 ± 1.49	1.40
50% ethanol	0.1	91.79 ± 4.89	5.33	89.59 ± 0.85	0.95
	0.5	110.38 ± 4.06	3.68	111.72 ± 3.36	3.01
	1	105.00 ± 3.21	4.77	104.89 ± 2.13	2.03
	5	112.94 ± 0.90	0.80	113.68 ± 1.71	1.51
	10	101.31 ± 0.19	0.19	101.98 ± 0.65	0.64
Liquid egg	0.1	99.49 ± 0.88	0.89	95.07 ± 4.37	4.59
	0.5	95.04 ± 3.55	5.84	89.08 ± 3.64	4.08
	1	93.57 ± 2.55	7.58	88.52 ± 2.18	3.57
	5	95.77 ± 4.07	7.34	90.66 ± 1.19	1.51
	10	91.70 ± 2.27	7.93	87.70 ± 1.33	1.52
Beef	0.1	93.02 ± 4.07	7.58	97.21 ± 1.00	11.32
	0.5	95.18 ± 2.25	2.36	98.09 ± 3.98	6.10
	1	97.60 ± 1.53	1.57	96.09 ± 3.54	3.68
	5	102.09 ± 1.27	1.25	104.69 ± 2.55	4.99
	10	89.53 ± 0.51	0.57	92.99 ± 4.43	4.77
Mackerel	0.1	92.06 ± 4.81	5.23	89.87 ± 1.33	1.48
	0.5	112.62 ± 4.14	3.68	113.99 ± 3.43	3.01
	1	105.00 ± 2.01	4.77	104.89 ± 2.13	2.03
	5	112.94 ± 0.90	0.80	113.68 ± 1.71	1.51
	10	101.31 ± 0.19	0.19	101.98 ± 0.65	0.64
Almond milk	0.1	98.18 ± 3.12	5.21	96.29 ± 2.48	5.69
	0.5	106.57 ± 1.02	0.96	107.98 ± 2.87	2.66
	1	110.83 ± 1.75	1.58	108.46 ± 4.29	4.88
	5	110.17 ± 1.82	1.65	110.65 ± 1.89	1.68
	10	98.69 ± 0.89	0.90	100.05 ± 2.15	2.15
Potato	0.1	104.40 ± 3.72	7.40	98.00 ± 4.81	4.90
	0.5	103.57 ± 1.24	1.20	106.83 ± 2.88	2.69
	1	93.19 ± 0.70	0.75	100.44 ± 2.88	8.18
	5	98.04 ± 0.20	0.21	102.79 ± 3.44	5.29
	10	83.11 ± 0.81	0.98	85.10 ± 2.13	2.50

**Table 5 foods-13-01888-t005:** Concentration of methylene blue in water and almond milk, determined independently in three laboratories.

	Concentration (μg/L (kg))	Detected Concentration in Other Laboratories (μg/L (kg))			Accuracy (%)
		Laboratory A	Laboratory B	Laboratory C	
Water	0.1	0.0919	0.1026	0.1015	98.66 ± 0.81
	1	0.9538	0.9568	0.9475	95.27 ± 0.26
	10	9.4133	9.9100	9.6852	96.67 ± 0.76
Almond milk	0.1	0.0894	0.1013	0.0952	95.28 ± 0.42
	1	0.8892	0.9522	0.9436	92.83 ± 0.71
	10	9.4117	9.9254	9.8979	97.45 ± 0.49

**Table 6 foods-13-01888-t006:** Quantification of methylene blue in printed TTIs by different experimental methods (elution test, transfer test, and full-scale transfer test).

TTI Type	Test Type	Sample	Concentration (μg/L (kg))
Printed TTI	Elution test	Water	0.06 ± 0.01
		4% acetic acid	1.20 ± 0.02
		n-heptane	0.19 ± 0.01
		20% ethanol	0.44 ± 0.03
		50% ethanol	0.10 ± 0.01
	Transfer test	Liquid egg	5.75 ± 0.01
		Beef	6.25 ± 0.01
		Mackerel	4.25 ± 0.01
		Almond milk	4.00 ± 0.02
		Potato	3.13 ± 0.01
	Full-scale transfer test	Liquid egg	8019.25 ± 4.29
		Beef	8820.75 ± 1.20
		Mackerel	11,745.00 ± 1.63
		Almond milk	4176.00 ± 2.09
		Potato	6097.25 ± 2.82

**Table 7 foods-13-01888-t007:** Quantification of methylene blue in Vitsab’s TTI using different experimental methods (elution test, transfer test, and full-scale transfer test).

TTI Type	Test Type	Sample	Concentration (μg/L (kg))
Vitsab’s TTI	Elution test	Water	0.05 ± 0.01
		4% acetic acid	0.18 ± 0.01
		n-heptane	ND ^a^
		20% ethanol	0.09 ± 0.01
		50% ethanol	0.08 ± 0.01
	Transfer test	Liquid egg	ND
		Beef	ND
		Mackerel	ND
		Almond milk	ND
		Potato	ND
	Full-scale transfer test	Liquid egg	12.00 ± 0.08
		Beef	6.50 ± 0.03
		Mackerel	7.50 ± 0.33
		Almond milk	4.75 ± 0.01
		Potato	4.50 ± 0.01

^a^ ND = not detected, the lower limit of detection.

## Data Availability

The original contributions presented in the study are included in the article, further inquiries can be directed to the corresponding author.

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
