# Peer review of "Evaluation of Methylene Blue Migration from Time—Temperature Indicators Using LC-MS/MS"

_foods, 2024, doi:10.3390/foods13121888_

Round 1
Reviewer 1 Report
Comments and Suggestions for Authors
A major drawback of this manuscript is that it cannot provide the enough significance of this work in the relevant field, as well as sufficient rationality of its experimental design. Anyway, it still might be reconsidered after undergoing major revisions.
1. The abstract should be rephrased as its current version is insufficient to provide detailed experimental methods and results, as well as a clear conclusion. The second paragraph seems to emphasize the experimental significance, but it seems too long to fit in the abstract.
2. The chemical structure of methylene blue is provided in Fig. 1. However, there is no label or description for this structure, thus it seems unrelated to this study. The authors should reveal whether this structure determines certain chemical properties of methylene blue, thereby affecting its migration.
3. While the manuscript discusses the regulatory landscape for methylene blue in various regions, it could benefit from a more detailed comparison and how the findings of this study could impact regulatory standards globally.
4. Migration analysis is a widely used technique. In the introduction, this manuscript does not introduce the current status of related research fields or the breakthroughs of this study on the basis of existing researches. In fact, it is difficult to claim that this work has “ developed an analytical method tailored to the food matrix group samples”. It seems only a minor patch on existing methods.
5. “Five kinds of food samples, including liquid egg, beef, mackerel, almond milk, and potato were purchased from the domestic market in the Republic of Korea.” For the food samples used in this study, more details should be provided, which is the most basic requirement for the reproducibility of a scientific work.
6. There is no explanation that these specific simulants (water, 4% acetic acid, n-hexane, 20% ethanol, and 50% ethanol) have sufficient representativeness for research. Food simulants were selected according to the guidelines of the South Korean Food and Drug Administration, which is just a local standard with very limited applications. The authors should use more authoritative standard such as AOAC-OMA or ISO, or even some publicly published research works.
7. A standard for dividing the food samples in this study is whether the protein content is greater than or less than 11%. The justification for the 11% protein content threshold is not clear and should be better explained.
8. The manuscript outlines different sample preparation methods for low and high protein content foods, but it does not clearly justify why these specific protocols were chosen or how they were optimized. It is confusing for replication purposes.
9. The study used varying conditions for different food simulants, such as heating at 70°C for some and 25°C for others. It lacks a comprehensive rationale for these conditions and their relevance to real-world scenarios.
10. The quality and clarity of Fig. 2 and Fig. 3 should be ensured. Labels for peaks and spectra details might be insufficient, and a better description of what these figures illustrate would be beneficial.
11. There are inconsistencies in terminology (e.g., "food simulant group" vs. "food matrix group") that can be standardized for clarity.
12. Some terms such as “declustering potential”, “entrance potential”, etc. should be used with necessary definitions or explanations.
13. There is no mention of ethical considerations or approvals, which are typically necessary for studies involving potentially hazardous materials.
14. Provide more detailed discussions on how the study’s findings can influence regulatory standards and what future research is needed.
Comments on the Quality of English LanguageModerate editing of English language required.
Author Response
Dear Reviewer 1
We appreciate your thorough review for the manuscript entitled “Evaluation of methylene blue migration from time temperature indicator using by LC-MS/MS”. We have carefully examined the comments and revised the manuscript accordingly as suggested. Please find the detailed responses below and the corresponding revisions/corrections highlighted/in track changes in the re-submitted files. We believe that the manuscript has been substantially improved and thank you for all the helpful comments to make this possible. We sincerely hope that the manuscript is now acceptable for the publication in Food Analytical Methods. If I can be of any further assistance in the review of the revised manuscript, please feel free to contact me at +82-31-961-5143 (Phone), or spartan@dongguk.edu (E-mail). Thank you again for your timely and highly constructive critiques.
Response to Comments and Suggestions
Comments 1: The abstract should be rephrased as its current version is insufficient to provide detailed experimental methods and results, as well as a clear conclusion. The second paragraph seems to emphasize the experimental significance, but it seems too long to fit in the abstract.
Response 1: We appreciate the Reviewer’s constructive comments. We agree with this comment. Therefore, the abstract was rewritten to be concise while including experimental methods, results, and clear conclusions. [The revised manuscript this change can be found – Page 1, paragraph 1, Line 10-21]
Comments 2: The chemical structure of methylene blue is provided in Fig. 1. However, there is no label or description for this structure, thus it seems unrelated to this study. The authors should reveal whether this structure determines certain chemical properties of methylene blue, thereby affecting its migration.
Response 2: We appreciate the Reviewer’s constructive comments. We agree with this comment. Therefore, added the structure and properties of methylene blue and added areas of use. [The revised manuscript this change can be found – Page 1, paragraph 4, Line 37-41]
Comments 3: While the manuscript discusses the regulatory landscape for methylene blue in various regions, it could benefit from a more detailed comparison and how the findings of this study could impact regulatory standards globally.
Response 3: We appreciate the Reviewer’s constructive comments. We agree with this comment. Therefore, added a more detailed comparison, considering revision request 3 and how it may impact regulatory standards globally. [The revised manuscript this change can be found – Page 2, paragraph 5, Line 55-65]
Comments 4: Migration analysis is a widely used technique. In the introduction, this manuscript does not introduce the current status of related research fields or the breakthroughs of this study on the basis of existing researches. In fact, it is difficult to claim that this work has “ developed an analytical method tailored to the food matrix group samples”. It seems only a minor patch on existing methods.
Response 4: We appreciate the Reviewer’s constructive comments. We agree with this comment. Previous studies on migration analysis have highlighted the role of variables like temperature and the shortcomings of using food simulants. We added that this study aims to provide more accurate data based on these results using real food samples and clarify its contribution to regulatory standards and deleted “develop”. [The revised manuscript this change can be found – Page 2, paragraph 7, Line 73-79]
Comments 5: “Five kinds of food samples, including liquid egg, beef, mackerel, almond milk, and potato were purchased from the domestic market in the Republic of Korea.” For the food samples used in this study, more details should be provided, which is the most basic requirement for the reproducibility of a scientific work.
Response 5: We appreciate the Reviewer’s constructive comments. We agree with this comment. Therefore, added details on food samples where to buy. [The revised manuscript this change can be found – Page 3, paragraph 11, Line 111-115]
Comments 6: There is no explanation that these specific simulants (water, 4% acetic acid, n-hexane, 20% ethanol, and 50% ethanol) have sufficient representativeness for research. Food simulants were selected according to the guidelines of the South Korean Food and Drug Administration, which is just a local standard with very limited applications. The authors should use more authoritative standard such as AOAC-OMA or ISO, or even some publicly published research works.
Response 6: We appreciate the Reviewer’s constructive comments. We agree with this comment. Therefore, he representativeness of food simulant group samples was further refined by adding cases from the United States and Europe. [The revised manuscript this change can be found – Page 3, paragraph 12, Line 118-132]
Comments 7: A standard for dividing the food samples in this study is whether the protein content is greater than or less than 11%. The justification for the 11% protein content threshold is not clear and should be better explained.
Response 7: We appreciate the Reviewer’s constructive comments. We agree with this comment. Therefore, It has been added that the standards for distinguishing between high-protein and low-protein foods are set in compliance with the standards established in accordance with Regulation (EU) No 1169/2011 and the food and labeling standards issued by the Ministry of Food and Drug Safety. [The revised manuscript this change can be found – Page 8, paragraph 31, Line 321-327]
Comments 8: The manuscript outlines different sample preparation methods for low and high protein content foods, but it does not clearly justify why these specific protocols were chosen or how they were optimized. It is confusing for replication purposes.
Response 8: We appreciate the Reviewer’s constructive comments. We agree with this comment. Therefore, reasons and explanations for why specific methods were selected have been added (Carrez, Hybrid SPE cartridge, Bond Elut Alumina A cartridge). [The revised manuscript this change can be found – Page 4, paragraph 17, Line 159-166]
Comments 9: The study used varying conditions for different food simulants, such as heating at 70°C for some and 25°C for others. It lacks a comprehensive rationale for these conditions and their relevance to real-world scenarios.
Response 9: We appreciate the Reviewer’s constructive comments. We agree with this comment. Therefore, added references from FDA and EFSA on temperature and time settings. [The revised manuscript this change can be found – Page 4, paragraph 19, Line 186-189]
Comments 10: The quality and clarity of Fig. 2 and Fig. 3 should be ensured. Labels for peaks and spectra details might be insufficient, and a better description of what these figures illustrate would be beneficial.
Response 10: We apologize for not guaranteeing the quality and clarity of fig 2 and fig 3. We agree with this comment. Therefore, ensures quality and clarity of Figures 2 and 3. [The revised manuscript this change can be found – Page 6, Line 237-243]
Comments 11: There are inconsistencies in terminology (e.g., "food simulant group" vs. "food matrix group") that can be standardized for clarity.
Response 11: We appreciate the Reviewer’s constructive comments. We agree with this comment. Therefore, added information about the need to classify samples into food simulant group and food matrix group to bridge the gap between laboratory tests and actual applications and obtain accurate and reliable results. [The revised manuscript this change can be found – Page 7, paragraph 27, Line 283-290]
Comments 12: Some terms such as “declustering potential”, “entrance potential”, etc. should be used with necessary definitions or explanations.
Response 12: We appreciate the Reviewer’s constructive comments. We agree with this comment. Therefore, added descriptions for DP, CE, EP, CEP, and CXP. [The revised manuscript this change can be found – Page 7, paragraph 22, Line 226-235]
Comments 13: There is no mention of ethical considerations or approvals, which are typically necessary for studies involving potentially hazardous materials.
Response 13: We appreciate the Reviewer’s constructive comments. We agree with this comment. Therefore, adds ethical considerations or approvals generally required for research involving potentially hazardous substances. [The revised manuscript this change can be found – Page 1, paragraph 4, Line 37-49]
Comments 14: Provide more detailed discussions on how the study’s findings can influence regulatory standards and what future research is needed.
Response 14: We appreciate the Reviewer’s constructive comments. We agree with this comment. Therefore, added how research findings may impact regulatory standards: Data on methylene blue can help establish safe limits for food contact materials and update packaging regulations.
Future research adds to the need to investigate long-term effects on a variety of foods and conditions using advanced methods to strengthen food safety standards. [The revised manuscript this change can be found – Page 14, paragraph 45-46, Line 493-506]
- Response to Comments on the Quality of English Language
Point 1: We appreciate the editor's feedback regarding the quality of the English language in our manuscript. We have carefully reviewed and revised the text to enhance clarity, coherence, and readability. Specifically, we have addressed the following issues:
- Corrected grammatical errors and awkward phrasing throughout the manuscript.
- Ensured consistent use of terminology and technical terms.
- Simplified complex sentences for better understanding.
- Enhanced the overall flow and structure of the text.
These revisions have been made to ensure that the manuscript meets the high standards expected by the journal and provides a clear and professional presentation of our research. We believe these changes significantly improve the quality of the English language in the manuscript. We welcome any further suggestions to improve the clarity and readability of our work.
Thank you for your constructive feedback.
- Additional clarifications
All references were reviewed, Table 8 was deleted and the contents were added to the main text.

Reviewer 2 Report
Comments and Suggestions for Authors
This study is interesting, however some comments are made.
If nothing was detected in table 8, it can be deleted and indicated in the text that it was not detected in this part.
The explanation of how methylene blue migrates with TTI to the food matrix is ​​a bit confusing. Although it is the validation of a method, not much is explained about the interactions that occur with migration and in the food matrix. I understand that it is a method to validate migration of toxic compounds, in this case methylene blue, but is this compound very common as a contaminant or toxic? We need to discuss this part more, or why do it based on this compound.
The differences between the different TTIs, advantages or disadvantages of each one or why study different ones are not well understood.
Reviewer 3 Report
Comments and Suggestions for Authors
The paper is well written and, in my opinion, it is suitable for pubblication in the present form.
I would like to suggest to the author to remove "intelligent packaging" from key words because "smart packaging" is sufficient. The explaination of the acronym LC-MS/MS in the key words is also non usefull beign it described in the abstract.
